# HSR-ENHANCED SPARSE ATTENTION ACCELERATION

## ABSTRACT

Large Language Models (LLMs) have demonstrated remarkable capabilities across various applications, but their performance on long-context tasks is often limited by the computational complexity of attention mechanisms. This paper introduces a novel approach to accelerate attention computation in LLMs, particularly for long-context scenarios. We leverage the inherent sparsity within attention mechanisms, both in conventional Softmax attention and ReLU attention (with ReLU$^\alpha$ activation, $\alpha \in \mathbb{N}_+$), to significantly reduce the running time complexity. Our method employs a Half-Space Reporting (HSR) data structure to rapidly identify non-zero or "massively activated" entries in the attention matrix. We present theoretical analyses for two key scenarios: attention generation and full attention computation with long input context. Our approach achieves a running time of $O(mn^{4/5})$ significantly faster than the naive approach $O(mn)$ for attention generation, where $n$ is the context length, $m$ is the query length, and $d$ is the hidden dimension. We can also reduce the running time of full attention computation from $O(mn)$ to $O(mn^{1-1/\lfloor d/2 \rfloor} + mn^{4/5})$. Importantly, our method introduces no error for ReLU attention and only provably negligible error for Softmax attention, where the latter is supported by our empirical validation. This work represents a significant step towards enabling efficient long-context processing in LLMs, potentially broadening their applicability across various domains.

## 1 INTRODUCTION

Large Language Models (LLMs) have showcased remarkable capabilities across various applications, including context-aware question answering, content generation, summarization, and dialogue systems, among others (Thoppilan et al., 2022; Coenen et al., 2021; Wei et al., 2022; Zhang et al., 2024b). Long-context tasks of LLMs have gained more and more attention. Several LLMs extend their context length to 128K tokens, such as Yarn (Peng et al., 2023), GPT-4 (OpenAI, 2023), Claude 3.5 (Anthropic, 2024), Llama 3.1 (Meta, 2024), Phi-3.5 (Abdin et al., 2024), Mistral Nemo (MistralAI, 2024), etc. A bottleneck for long-context tasks is the computational cost of the attention mechanism in LLMs. The key to LLM success is the transformer architecture (Vaswani et al., 2017), wildly used in various practical scenarios (Radford et al., 2019; Kenton & Toutanova, 2019; Wang et al., 2023b;a; 2024), whose critical component is the attention mechanism. Let $n$ be the data length, $m$ be the length of query tokens, and $d$ be the feature dimension[1]. The conventional attention uses Softmax activation and is defined as follows:

**Definition 1.1** (Softmax attention). *Let $Q \in \mathbb{R}^{m \times d}$ and $K, V \in \mathbb{R}^{n \times d}$ denote the query, key, and value matrix. The Softmax attention is:*

$$\mathsf{Attn}_s(Q, K, V) := \mathsf{Softmax}(QK^\top)V = D^{-1}A_s V \in \mathbb{R}^{m \times d},$$

*where (1) $A_s := \exp(QK^\top/\sqrt{d}) \in \mathbb{R}^{m \times n}$ and $\exp$ is applied element-wise , (2) $D := \mathrm{diag}(A_s \cdot \mathbf{1}_n) \in \mathbb{R}^{m \times m}$ denotes the normalization matrix, (3) $D^{-1}A_s \in \mathbb{R}^{m \times n}$ denotes the attention matrix.*

In practical LLM applications, there are two scenarios for attention computation depending on the context length $n$ and query length $m$. The first case, $m = \Theta(1)$, represents the iterative text generation based on the pre-computed Key Value Cache (KV), which stores the intermediate attention

---

[1]As $d$ is always fixed in practice, there is no need to scale up $d$ in analysis. Thus, in this work, we always assume $d$ is a small constant.

key and value matrices. The second case, $m = \Theta(n)$, represents the full self-attention computation before text generation or the cross-attention computation. However, in both cases, when the context window $n$ becomes larger, the running time will increase correspondingly, i.e., it will be linear and quadratic in $n$ for $m = \Theta(1)$ and $m = \Theta(n)$, respectively. Thus, reducing the running time of attention computations with long context input becomes essential to minimize response latency and increase throughput for LLM API calls.

In this work, we introduce novel methods to reduce the running time complexity for both cases, i.e., $m = \Theta(1)$ and $m = \Theta(n)$. Our approach is inspired by the inherent sparsity found within attention mechanisms. Numerous prior studies have highlighted the significant sparsity in the attention matrix (Child et al., 2019; Anagnostidis et al., 2023; Liu et al., 2023; Tang et al., 2024; Sun et al., 2024). This manifestation of sparsity in Softmax attention is that a large number of attention scores, i.e., $QK^\top$, concentrate on a small number of entries, which is known as "massive activation". Due to this nature, Softmax attention can be accelerated by only calculating the entries that contain large attention scores, introducing negligible approximation errors (Zhang et al., 2023; Li et al., 2024).

Moreover, when considering ReLU attention (with ReLU$^\alpha$ activation, $\alpha \in \mathbb{N}_+$), we can accelerate the attention computation *without* any approximation error. ReLU attention is another attention mechanism used in transformer architecture, substituting the conventional Softmax activation function with ReLU, which has demonstrated performance comparable to Softmax attention in various downstream tasks (Wortsman et al., 2023; Hua et al., 2022); see Section 2 for more details. In the following, we present the formal definition of ReLU attention.

**Definition 1.2** (ReLU attention). *Let $Q \in \mathbb{R}^{m \times d}$ and $K, V \in \mathbb{R}^{n \times d}$ denote the query, key, and value matrix. Let $\alpha \in \mathbb{N}_+$. The ReLU attention is:*

$$\mathsf{Attn}_r(Q, K, V) := D^{-1} A_r V \in \mathbb{R}^{m \times d},$$

*where (1) $A_r := \mathsf{ReLU}^\alpha(QK^\top/\sqrt{d} - b) \in \mathbb{R}^{m \times n}$ and $\mathsf{ReLU}^\alpha$ denotes the $\alpha$-th power of ReLU activation for any $\alpha \in \mathbb{N}_+$, (2) $D := \mathrm{diag}(A_r \cdot \mathbf{1}_n) \in \mathbb{R}^{m \times m}$ denotes the normalization matrix, (3) $b \in \mathbb{R}$ denotes position bias, (4) $D^{-1} A_r \in \mathbb{R}^{m \times n}$ denotes the attention matrix.*

To expedite the computation, the critical task is to identify the large/non-zero entries for Softmax/ReLU attention, respectively. To do so, we utilize the half-space reporting (HSR) data structure, which is introduced in Agarwal et al. (1992) to address the half-space range reporting problem. This is a fundamental problem in computational geometry and can be formally defined as follows:

**Definition 1.3** (Half-space range reporting (Agarwal et al., 1992; Song et al., 2021)). *Given a set $S$ of $n$ points in $\mathbb{R}^d$ with initialization, we have an operation $\mathrm{QUERY}(H)$: given a half-space $H \subset \mathbb{R}^d$, output all of the points in $S$ that contain in $H$, i.e., $S \cap H$.*

In our framework, we define the half-space as the region where the attention scores (the inner products of key and query vectors) exceed some threshold. We leverage this data structure to expedite the identification of non-zero entries within the ReLU attention matrix and large en-

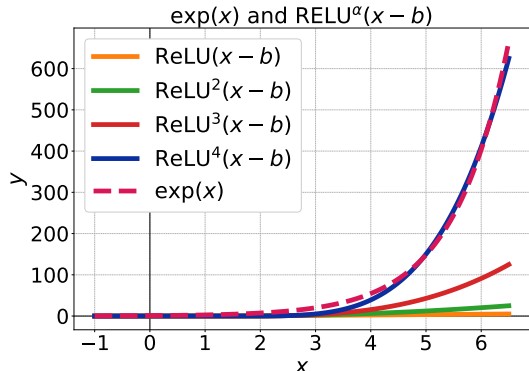

Figure 1: The trending of the Softmax activation (exp) and the ReLU activation with different powers. Here, we choose $b = 1.5$ as the threshold for the ReLU activation.

tries in Softmax attention. Consequently, we can compute the ReLU attention only based on those non-zero entries without any approximation error, and compute the Softmax attention based on entries larger than threshold with negligible approximation errors, resulting in a substantial reduction in computation time. When $m = \Theta(1)$, our methods can significantly accelerate ReLU and Softmax attention computation time over the naive approach from $O(mn)$ to $O(mn^{4/5})$ with pre-processed KV cache. When $m = \Theta(n)$, our online methods can also accelerate ReLU and Softmax attention computation time over the naive approach from $O(mn)$ to $O(mn^{1-1/\lfloor d/2 \rfloor} + mn^{4/5})$. In more

details, when $m = \Theta(1)$ and for any $d \in \mathbb{N}_+$, our Algorithm 2 can achieve the fast generation with pre-processed KV cache in $O(mn^{4/5})$ (Theorem 4.1 and Theorem 4.2)). When $m = \Theta(n)$, our Algorithm 3 can achieve the full attention computation in $O(mn^{1-1/\lfloor d/2 \rfloor} + mn^{4/5})$ including HSR initialization time and query time (Theorem 5.1 and Theorem 5.2). Thus, our methods can improve both the generation speed and full attention computation for long input context, i.e., $n$ being excessively large. Furthermore, our empirical results in Section 7 show that the approximation error associated with Softmax attention utilizing "massive activated" entries only is small in practice, which is consistent with our theoretical analysis.

**Our contributions:**

- To the best of our knowledge, this is the first work incorporating the HSR data structure with attention computation, to reduce the running time complexity with the help of the sparsity within the attention mechanisms.

- Theoretically, we provide rigorous proofs for reducing the computational time (1) for ReLU attention generation from $O(mn)$ to $O(mn^{4/5})$ (Algorithm 2 and Theorem 4.1); (2) for full ReLU attention computation from $O(mn)$ to $O(mn^{1-1/\lfloor d/2 \rfloor} + mn^{4/5})$ (Algorithm 3 and Theorem 5.1), without incurring any approximation error in both cases.

- We achieve the same running time speed up for the conventional Softmax attention, and we give rigorous theoretical proofs to ensure that the resulting approximation error remains negligible (Theorem 4.2, 5.2 and Theorem 4.3).

- We conduct empirical experiments on prominent LLMs to verify the approximation error associated with Softmax attention utilizing "massive activated" entries only. The results show that the error using a few top entries is already insignificant, consistent with our theoretical analysis.

**Roadmap.** Section 2 presents related work. In Section 3, we introduce essential concepts and key definitions used this paper. In Section 4, we present our main results, i.e., guarantees on run time reduction and approximation error. In Section 5, we introduce the extension of our method on full attention computation. In Section 6, we provide a brief summary of the techniques used in our proof. In Section 7, we provide our empirical results of evaluating three mainstream LLMs with Softmax attention with top-$r$ indices on different $r$. In Section 8, we discuss the potential of extending our method to other activation functions. In Section 9, we concludes our algorithm and contributions.

## 2 RELATED WORK

**Attention acceleration for long context input.** Long context window is essential for transformer based LLMs in many downstream tasks. However, due to the quadratic time complexity associated with self-attention mechanisms, transformers are usually hard to inference efficiently. Numerous methods have been proposed to enhance the inference efficiency. One approach involves using alternative architectures as proxies for attention to support faster inference, such as Mamba (Gu & Dao, 2023; Dao & Gu, 2024), PolySketchFormer (Kacham et al., 2023), and Linearizing Transformers (Zhang et al., 2024a; Mercat et al., 2024). However, the broad applicability of these methods across different applications and modalities remains to be fully validated. Another line of research focuses on approximating attention matrix computation (Alman & Song, 2023; 2024a;b; Han et al., 2024; Zandieh et al., 2024; Liang et al., 2024d; Poli et al., 2023; Cai et al., 2024; Liang et al., 2024c;a; Gao et al., 2023; Dong et al., 2024; Liang et al., 2024b). Nevertheless, these methods often rely on assumptions that may not be practical. For instance, some approaches use polynomial methods to approximate the exponential function, which requires all entries to be bounded by a small constant. However, our HSR-enhanced attention framework is designed based on practical observation and validated by empirical support.

**ReLU attention.** ReLU attention is an innovative mechanism that employs the ReLU activation function in place of the traditional Softmax function for attention computation. Previous studies have highlighted the promise potential of ReLU attention in various domains. From empirical side, Wortsman et al. (2023) has demonstrated that incorporating ReLU as the activation function in vision transformers enhances performance on downstream tasks. Shen et al. (2023) has shown that

transformers equipped with ReLU attention outperform those with Softmax attention, particularly when dealing with large key-value memory in machine translation tasks. From theoretical side, the scale-invariant property of ReLU attention (Li et al., 2022) facilitates the scalability of transformer networks. Furthermore, Bai et al. (2023); Fu et al. (2023) have shown that the inherent properties of ReLU attention contribute positively to the learning process of transformer models. Another key advantage of ReLU attention is that the ReLU function effectively sets all negative values to zero, allowing us to bypass these non-contributory elements during attention computation, thereby reducing the running time of attention computation. Importantly, omitting these zero and negative entries does not introduce any error into the final output of the ReLU attention mechanism.

**Half-space reporting (HSR) data structure.** The Half-Space Reporting (HSR) data structure, initially proposed by Agarwal et al. (1992), was developed to address the half-space range reporting problem. The expedited range query capability inherent to HSR has been demonstrated to significantly enhance computational efficiency across a variety of tasks, as evidenced by numerous previous works in the literature. Studies such as Jiang et al. (2021) and Bhattacharya et al. (2023) have applied HSR to facilitate solving general linear programming (LP) problems. Another line of research has highlighted HSR's potential in expediting the training process of contemporary neural networks (Qin et al., 2023; Gao et al., 2022). There is also a collection of research that concentrates on leveraging HSR for the advancement of solutions to geometric and graphical challenges (Chen et al., 2005; Ju et al., 2013; Eppstein et al., 2017).

## 3 PRELIMINARY

In Section 3.1, we introduce notations used in the paper. In Section 3.2, we introduce a modified version of Softmax attention that operates on a specific subset of indices. It defines the top-$r$ nearest neighbors Softmax attention, which focuses on the most relevant entries in the attention matrix. In Section 3.3, we describe the massive activation property for attention mechanisms. In Section 3.4, we present a data structure for efficiently solving the half-space range reporting problem.

### 3.1 NOTATIONS

Here, we introduce basic notations used in this paper. For any positive integer $n$, we use $[n]$ to denote set $\{1, 2, \cdots, n\}$. We use $\mathrm{Var}[]$ to denote the variance. For two vectors $x \in \mathbb{R}^n$ and $y \in \mathbb{R}^n$, we use $\langle x, y \rangle$ to denote the inner product between $x, y$. We use $\mathbf{1}_n$ to denote a length-$n$ vector where all the entries are ones. We use $X_{i,j}$ to denote the $i$-row, $j$-th column of $X \in \mathbb{R}^{m \times n}$. We use $\|A\|_\infty$ to denote the $\ell_\infty$ norm of a matrix $A \in \mathbb{R}^{n \times d}$, i.e. $\|A\|_\infty := \max_{i \in [n], j \in [d]} |A_{i,j}|$.

### 3.2 SOFTMAX ATTENTION WITH INDEX SET

Recall that we have already provided the definition of ReLU attention in Definition 1.2. Here, we present the key concepts of Softmax attention. For Softmax attention, since we only calculate the "massive activated" entries to get our approximated results, we introduce the formal definition:

**Definition 3.1** (Input with index set). *Let $K \in \mathbb{R}^{n \times d}$ and $V \in \mathbb{R}^{n \times d}$ be defined in Definition 1.1. Let $R \subseteq [n]$ be an index set of size $|R| = r \in [n]$. Let $\overline{R} := [n] \setminus R$ be the complementary set, where $|\overline{R}| = n - r$. We define*

$$\widehat{K} := K_R \in \mathbb{R}^{r \times d} \quad \widehat{V} := V_R \in \mathbb{R}^{r \times d} \quad \overline{K} := K_{\overline{R}} \in \mathbb{R}^{(n-r) \times d} \quad \overline{V} := V_{\overline{R}} \in \mathbb{R}^{(n-r) \times d}$$

*as the submatrix of $K$ and $V$, i.e., whose row index is in $R$ or $\overline{R}$, respectively.*

In this work, we consider calculating the Softmax attention on the "massive activation" index set, where we define the "massive activation" index set as the top-$r$ indices. We introduce our definition for top-$r$ indices of Softmax attention as follows:

**Definition 3.2** ( Top-$r$ indices Softmax attention ). *Let $q \in \mathbb{R}^d$, $K, V \in \mathbb{R}^{n \times d}$ be defined in Definition 1.1. Let $\mathsf{NN}(r, q, K) \subseteq [n]$ denote the indices of top-$r$ entries of $qK$, where $|\mathsf{NN}(r, q, K)| = r$. Let $\widehat{K}, \widehat{V} \in \mathbb{R}^{r \times d}$ and $\overline{K}, \overline{V} \in \mathbb{R}^{(n-r) \times d}$ be defined in Definition 3.1. We define the top-$r$ nearest neighbors (NN) Softmax attention computation $\widehat{\mathsf{Attn}}_s(q, K, V) \in \mathbb{R}^d$ as follows:*

$$\widehat{\mathsf{Attn}}_s(q, K, V) := \mathsf{Softmax}(q\widehat{K}^\top)\widehat{V} = \widehat{\alpha}^{-1}\widehat{u}\widehat{V} \in \mathbb{R}^d$$

*where*

$$\widehat{u} := \exp(q\widehat{K}^\top) \in \mathbb{R}^r \quad and \quad \widehat{\alpha} := \langle \widehat{u}, \mathbf{1}_r \rangle \in \mathbb{R}.$$

*Furthermore, we define* $\overline{u} := \exp(q\overline{K}^\top) \in \mathbb{R}^{n-r}$, $\overline{\alpha} := \langle \overline{u}, \mathbf{1}_{n-r} \rangle \in \mathbb{R}$, *and* $u := \exp(qK^\top) \in \mathbb{R}^{n+1}$, $\alpha := \langle u, \mathbf{1}_{n+1} \rangle \in \mathbb{R}$.

In Definition 3.2, we view the "massive activated" entries as the top-$r$ entries. Therefore, we only calculate the Softmax attention based on $\widehat{K}, \widehat{V} \in \mathbb{R}^{r \times d}$, instead of $K, V \in \mathbb{R}^{n \times d}$.

### 3.3 MASSIVE ACTIVATION

Now, we introduce our observations on the properties of the attention scores (the inner products of query vectors and key vectors). This further facilitates the error analysis of the top-$r$ indices Softmax attention. To begin with, we provide the definition of the massive activation property as follows:

**Definition 3.3** (Massive activation property)**.** *Let* $\gamma \in [0, 1]$, $\beta_1 \geq \beta_2 \geq 0$. *Let* $\mathsf{NN}(r, q, K) \subseteq [n]$ *denote the indices of top-r entries of* $qK$. *We define* $(\gamma, \beta_1, \beta_2)$ *massive activation for a query* $q \in \mathbb{R}^d$ *and key cache* $K \in \mathbb{R}^{n \times d}$, *if the following conditions hold:*

- *The top-$n^\gamma$ entries are massive, i.e.,* $\frac{1}{n^\gamma \cdot \|q\|_2} \sum_{i \in \mathsf{NN}(n^\gamma, q, K)} \langle q, K_i \rangle \geq \beta_1 \log(n)$.

- *The remaining terms are upper bounded, i.e,* $\forall i \in [n] \setminus \mathsf{NN}(n^\gamma, q, K)$, $\frac{1}{\|q\|_2} \langle q, K_i \rangle \leq \beta_2 \log(n)$.

An intuitive understanding of Definition 3.3 is that, the summation of "massive activated" entries dominates the summation of all entries, and the entries we ignored only contributes little to the final summation. Therefore, it is reasonable for us to omit those non "massive activated" entries.

**Remark 3.4.** *There are many distributions satisfying the property in Definition 3.3, such as (1) K drawing from any subexponential distribution, e.g., multivariate Laplace distributions, (2) K drawing from any mixture of Gaussian distribution with* $n^{1-\gamma}$ *Gaussian clusters.*

### 3.4 HALF-SPACE REPORTING (HSR) DATA STRUCTURE

---

**Algorithm 1** Half Space Report Data Structure

---

1: **data structure** HALFSPACEREPORT
2:       INIT$(S, n, d)$            ▷ Initialize the data structure with a set $S$ of $n$ points in $\mathbb{R}^d$
3:       QUERY$(a, b)$            ▷ $a, b \in \mathbb{R}^d$. Output the set $\{x \in S : \mathrm{sgn}(\langle a, x \rangle - b) \geq 0\}$
4: **end data structure**

---

We restate the result from Agarwal et al. (1992) for solving the half-space range reporting problem. The interface of their algortihm can be summarized as in Algorithm 1. Intuitively, the data-structure recursively partitions the set $S$ and organizes the points in a tree data-structure. Then for a given query $(a, b)$, all $k$ points of $S$ with $\mathrm{sgn}(\langle a, x \rangle - b) \geq 0$ are reported quickly. Note that the query $(a, b)$ here defines the half-space $H$ in Definition 1.3. We summarize the time complexity of HSR data structure as follows:

**Corollary 3.5** (HSR data-structure time complexity Agarwal et al. (1992), informal version of Corollary A.7)**.** *Let* $\mathcal{T}_{\mathsf{init}}$ *denote the pre-processing time to build the data structure,* $\mathcal{T}_{\mathsf{query}}$ *denote the time per query and* $\mathcal{T}_{\mathsf{update}}$ *time per update. Given a set of n points in* $\mathbb{R}^d$, *the half-space range reporting problem can be solved with the following performances:*

- *Part 1.* $\mathcal{T}_{\mathsf{init}}(n, d) = O_d(n \log n)$, $\mathcal{T}_{\mathsf{query}}(n, d, k) = O(dn^{1-1/\lfloor d/2 \rfloor} + dk)$.

- *Part 2.* $\mathcal{T}_{\mathsf{init}}(n, d) = O(n^{\lfloor d/2 \rfloor})$, $\mathcal{T}_{\mathsf{query}}(n, d, k) = O(d \log(n) + dk)$.

## 4 MAIN RESULTS ON ATTENTION GENERATION

In this section, we present our key findings regarding attention generation, $m = \Theta(1)$, for both ReLU and Softmax attention mechanisms. Across both scenarios, we have reduced the time complexity from a naive $O(mn)$ to $O(mn^{4/5})$. Specifically, for the ReLU attention model, we have

managed to accelerate the processing time without introducing any approximation errors. In the case of Softmax attention, our technique results in only an insignificant approximation error.

---

**Algorithm 2** Attention generation

---

1: **data structure** ATTENTIONGENERATION                    ▷ Lemma 6.2
2: **members**
3:    HALFSPACEREPORT HSR                    ▷ Algorithm 1, Part 2 of Corollary 3.5
4:    $\{K_i\}_{i \in [n]}$                    ▷ Key matrix
5:    $V \in \mathbb{R}^{n \times d}$                    ▷ Value matrix
6:    $b \in \mathbb{R}$                    ▷ Threshold of ReLU activation
7: **end members**
8: **procedure** INIT($\{K_i\}_{i \in [n]}, V, n, d$)
9:    $\{K_i\}_{i \in [n]}, V \leftarrow \{K_i\}_{i \in [n]}, V$                    ▷ Store necessary matrices
10:    $b \leftarrow \sigma_a \cdot \sqrt{0.4 \log n}$            ▷ Init essential parameters and data structure. Lemma 6.1
11:    HSR.INIT($\{K_i\}_{i \in [n]}, n, d$)                    ▷ It takes $\mathcal{T}_{\mathsf{init}}(n, d)$ time
12: **end procedure**
13: **procedure** INFERENCE($Q \in \mathbb{R}^{m \times d}, m$)
14:    $A \leftarrow \mathbf{0}_{m \times n}$
15:    **for** $i = 1 \rightarrow m$ **do**                    ▷ Loop for $m$ query vectors
16:        $\widetilde{S}_{i,\mathrm{fire}} \leftarrow$ HSR.QUERY($Q_i, b$)                    ▷ It takes $\mathcal{T}_{\mathsf{query}}(n, d, \widetilde{k}_i)$ time
17:        **for** $j \in \widetilde{S}_{i,\mathrm{fire}}$ **do**            ▷ Calculate the ReLU attention output according to $\widetilde{S}_{i,\mathrm{fire}}$
18:            $A_{i,j} \leftarrow \mathsf{ReLU}^\alpha(\langle Q_i, K_j\rangle/\sqrt{d} - b)$ or $A_{i,j} \leftarrow \mathsf{Softmax}(\langle Q_i, K_j\rangle/\sqrt{d})$
19:        **end for**
20:    **end for**
21:    **return** $D^{-1}AV$
22: **end procedure**
23: **end data structure**

---

We begin with introducing our result on ReLU attention generation as follows:

**Theorem 4.1** (Running time of ReLU attention generation, informal version of Theorem C.2). *Let ReLU attention be defined as Definition 1.2. Assume each entry of $K$ is from Gaussian $\mathcal{N}(0, \sigma_k^2)$, and each entry of $Q$ is from Gaussian $\mathcal{N}(0, \sigma_q^2)$. Let $\delta \in (0, 1)$ denote the failure probability. Let $\sigma_a = 4 \cdot (1 + d^{-1} \log(m/\delta))^{1/2} \cdot \sigma_q \sigma_k$. Let $b = \sigma_a \cdot \sqrt{0.4 \log n}$. Suppose we have KV Cache $K, V \in \mathbb{R}^{n \times d}$. We want to generate a $m$ length answer, where $n \gg m$. Then, our inference function in Algorithm 2, with probability at least $1 - \delta$, takes $O(mn^{4/5})$ time to generate the answer.*

Theorem 4.1 shows that our Algorithm 2 accelerates the running time of ReLU attention generation from naive $O(mn)$ to $O(mn^{4/5})$, which is a significant speed up when the KV Cache is large. The at least $1 - \delta$ success probability originates from the sparsity analysis of ReLU attention (Lemma 6.1), where with probability at least $1 - \delta$, we have the number of non-zero entries of each row of the attention matrix is at most $n^{4/5}$.

Then, we move on to presenting our result on Softmax attention generation. Our results consist two parts: the improved running time of Softmax attention generation, and the error analysis of Softmax attention with index set. Firstly, we introduce our result about the imporved running time of Softmax attention generation as follows:

**Theorem 4.2** (Running time of Softmax attention generation, informal version of Theorem E.1). *Let $Q \in \mathbb{R}^{m \times d}$, $K, V \in \mathbb{R}^{n \times d}$ and the Softmax attention $\mathsf{Attn}_s$ be defined in Definition 1.1. Let $\mathsf{NN}(r, q, K) \subseteq [n]$ and the Softmax attention with index set $\widehat{\mathsf{Attn}}_s$ be defined as Definition 3.2. We choose the threshold $b \in \mathbb{R}$ in Algorithm 2 such that $R = \mathsf{NN}(n^{4/5}, q, K)$. Then, we can show that the Softmax attention with index set $\widehat{\mathsf{Attn}}_s$ achieves outstanding running time under the Softmax attention generation scenario: Suppose we have KV Cache $K, V \in \mathbb{R}^{n \times d}$. We want to generate a $m$ length answer, where $n \gg m$. Our inference function in Algorithm 2 (replacing ReLU attention with Softmax attention) takes $O(mn^{4/5})$ time to generate the answer.*

Theorem 4.2 demonstrates that if we choose the threshold $b$ satisfying $R = \mathsf{NN}(n^{4/5}, q, K)$, we can achieve a significant running time improve of the Softmax attention generation.

It is evident that this method introduces an approximation error due to the exclusion of certain entries. Nevertheless, under mild assumptions about the distribution of the attention scores, we demonstrate that this approximation error is indeed negligible. The proof's intuitive explanation lies in the fact that the majority of attention scores are focused on the small subset of entries that we retain. We organize our result as follows:

**Theorem 4.3** (Error analysis of Softmax attention with index set, informal version of Theorem F.2)**.** *Let $Q \in \mathbb{R}^{m \times d}$, $K, V \in \mathbb{R}^{n \times d}$ and the Softmax attention $\mathsf{Attn}_s$ be defined in Definition 1.1. Let $q \in \mathbb{R}^d$ denote a single row of $Q \in \mathbb{R}^{m \times d}$. Let $\gamma \in [0, 1]$, $\beta_1 \geq \beta_2 \geq 0$. Let the index set $R$ and the Softmax attention with index set $\widehat{\mathsf{Attn}}_s$ be defined as Definition 3.2. Let $\mathsf{NN}(r, q, K) \subseteq [n]$ denote the indices of top-r entries of $qK$. Let $R = \mathsf{NN}(n^\gamma, q, K) \subseteq [n]$, where $|R| = n^\gamma$. Assume the query $q$ and key cache $K$ have $(\gamma, \beta_1, \beta_2)$ massive activation property (Definition 3.3). Then, we have*

$$\|\widehat{\mathsf{Attn}}_s(q, K, V) - \mathsf{Attn}_s(q, K, V)\|_\infty \leq \frac{2\|V\|_\infty}{n^{\gamma + (\beta_1 - \beta_2) \cdot \|q\|_2 - 1}}.$$

Theorem 4.3 presents the error of Softmax attention with index set is relatively small. Consequently, omitting the remaining less significant entries is a justifiable compromise.

**Remark 4.4.** *With mild assumptions on $V$, we can have more precious results from Theorem 4.3. For example, if the entries in $V$ conform to subgaussian distribution with constant variance, we have $\|V\|_\infty = O(\log(n))$ with high probability.*

## 5 EXTENSION ON FULL ATTENTION COMPUTATION

In this section, we extend our results to full attention computation scenario, where the number of queries and keys is proportional, i.e., $m = \Theta(n)$. Essentially, the full attention computation is beneficial in practical applications, particularly within the context of cross-attention computations. For ReLU attention, we leverage Part 1 result of Corollary 3.5 to accelerate the identification of non-zero entries (activated entries). We introduce our result on ReLU attention as follows:

---

**Algorithm 3** Full attention computation

1: **data structure** FULLATTENTIONCOMPUTATION        ▷ Lemma 6.3
2: **members**
3:   HALFSPACEREPORT HSR       ▷ Algorithm 1, Part 1 of Corollary 3.5
4: **end members**
5:
6: **procedure** INFERENCE($\{K_i\}_{i \in [n]}, \{Q_r\}_{r \in [m]}, V, n, m, d$)
7:   $b \leftarrow \sigma_a \cdot \sqrt{0.4 \log n}$.      ▷ Threshold of ReLU activation (Lemma 6.1)
8:   HSR.INIT($\{K_i\}_{i \in [n]}, n, d$)        ▷ It takes $\mathcal{T}_{\mathsf{init}}(n, d)$ time
9:   $A \leftarrow \mathbf{0}_{m \times n}$
10:   **for** $i = 1 \to m$ **do**         ▷ Loop for $m$ query vectors
11:    $\widetilde{S}_{i,\mathsf{fire}} \leftarrow$ HSR.QUERY($Q_i, b$)      ▷ It takes $\mathcal{T}_{\mathsf{query}}(n, d, \widetilde{k}_i)$ time.
12:    **for** $j \in \widetilde{S}_{i,\mathsf{fire}}$ **do**    ▷ Calculate the ReLU attention output according to $\widetilde{S}_{i,\mathsf{fire}}$
13:     $A_{i,j} \leftarrow \mathsf{ReLU}^\alpha(\langle Q_i, K_j\rangle / \sqrt{d} - b)$ or $A_{i,j} \leftarrow \mathsf{Softmax}(\langle Q_i, K_j\rangle / \sqrt{d})$
14:    **end for**
15:   **end for**
16:   **return** $D^{-1}AV$
17: **end procedure**
18: **end data structure**

---

**Theorem 5.1** (Running time of full ReLU attention computation, informal version of Theorem B.2)**.** *Let ReLU attention be defined as Definition 1.2. Assume each entry of $K$ is from Gaussian $\mathcal{N}(0, \sigma_k^2)$, and each entry of $Q$ is from Gaussian $\mathcal{N}(0, \sigma_q^2)$. Let $\delta \in (0, 1)$ denote the failure probability. Let $\sigma_a = 4 \cdot (1 + d^{-1}\log(m/\delta))^{1/2} \cdot \sigma_q \sigma_k$. Let $b = \sigma_a \cdot \sqrt{0.4 \log n}$. Suppose we have $Q, K, V \in \mathbb{R}^{n \times d}$. There exist an algorithm (Algorithm 3), with probability at least $1 - \delta$, takes $O(n^{2-1/\lfloor d/2 \rfloor} + n^{1+4/5})$ time to compute the full ReLU attention of $Q, K, V$.*

In Theorem 5.1, we improve the running time of full ReLU attention computation from $O(n^2)$ to $O(n^{2-1/\lfloor d/2 \rfloor} + n^{1+4/5})$, which is a notable uplift of the running time when $n$ is extremely large.

Then, we present our result on Softmax attention. Intuitively, we use the Part 1 result of Corollary 3.5 to identify those "massive activated" entries (top-$r$ indices) within the attention matrix of Softmax attention, and calculate the Softmax attention with top-$r$ indices. We organize our result as follows:

**Theorem 5.2** (Running time of Softmax full attention computation, informal version of Theorem E.2). *Let $Q \in \mathbb{R}^{m \times d}$, $K, V \in \mathbb{R}^{n \times d}$ and the Softmax attention $\mathsf{Attn}_s$ be defined in Definition 1.1. Let $\mathsf{NN}(r, q, K) \subseteq [n]$ and the Softmax attention with index set $\widehat{\mathsf{Attn}}_s$ be defined as Definition 3.2. We choose the threshold $b \in \mathbb{R}$ in Algorithm 3 such that $R = \mathsf{NN}(n^{4/5}, q, K)$. Then, we have the Softmax attention with index set $\widehat{\mathsf{Attn}}_s$ achieves outstanding running time under full Softmax attention computation scenario: Suppose we have $m = \Theta(n)$. Algorithm 3 (replacing ReLU attention with Softmax attention) takes $O(n^{2-1/\lfloor d/2 \rfloor} + n^{1+4/5})$ time to compute the full ReLU attention of $Q, K, V$.*

Theorem 5.2 demonstrates our $O(n^{2-1/\lfloor d/2 \rfloor} + n^{1+4/5})$ running time on Softmax full attention computation, which improves from naive running time $O(n^2)$.

# 6 TECHNICAL OVERVIEW

In Section 6.1, we introduce our analysis about the sparsity in the ReLU attention mechanism. In Section 6.2, we present our results of two general attention frameworks. In Section 6.3, we provide our error analysis of Softmax attention with index set. We have shown that with mild assumption on the distribution of attention scores, the error of Softmax attention with index set is negligible.

## 6.1 SPARSITY ANALYSIS OF RELU ATTENTION

Intuitively, the ReLU activation will deactivate some key and query pairs. We introduce the results of employing the concentration inequality to quantitatively analyze the number of non-zero entries.

**Lemma 6.1** (Sparsity analysis, informal version of Lemma D.3). *Let the ReLU attention be defined as Definition 1.2. Let $Q \in \mathbb{R}^{m \times d}$ and $K, V \in \mathbb{R}^{n \times d}$ be defined as Definition 1.2. Let $b \in \mathbb{R}$ denote the threshold of ReLU activation, as defined in Definition 1.2. For $i \in [m]$, let $\widetilde{k}_i$ denote the number of non-zero entries in $i$-th row of $A \in \mathbb{R}^{m \times n}$. Assume each entry of $K$ is from Gaussian $\mathcal{N}(0, \sigma_k^2)$, and each entry of $Q$ is from Gaussian $\mathcal{N}(0, \sigma_q^2)$. Let $\delta \in (0, 1)$ denote the failure probability. Let $\sigma_a = 4 \cdot (1 + d^{-1} \log(m/\delta))^{1/2} \cdot \sigma_q \sigma_k$. Let $b = \sigma_a \cdot \sqrt{0.4 \log n}$. Then, we have, with probability at least $1 - \delta$, for all $i \in [m]$, the number of non-zero entries of the $i$-th row $\widetilde{k}_i$ is at most $2n^{4/5}$.*

In Lemma 6.1, we use $\widetilde{k}_i$ to denote the number of non-zero entries in $i$-th row of attention matrix $A_r \in \mathbb{R}^{m \times n}$. It indicates that if we choose $b = \sigma_a \sqrt{0.4 \log n}$, with high probability, the number of activated (non-zero) entries can be bounded by $O(n^{4/5})$.

## 6.2 GENERAL ATTENTION FRAMEWORKS

First, we introduce our general framework for attention generation computation. Here, we use the Part 1 result of the HSR data structure. As for this framework is designed for the attention generation task, the key matrix $K$ is fixed in each inference. Therefore, in the INIT procedure, we initialize the HSR data structure with the key matrix $K$. Then, in each inference, we use the same HSR data structure to answer the query from each row of the query matrix $Q$. We provide the result of this general attention generation framework as follows.

**Lemma 6.2** (General attention generation framework, informal version of Lemma C.1). *Let $Q \in \mathbb{R}^{m \times d}$ and $K, V \in \mathbb{R}^{n \times d}$ be defined as Definition 1.2. Assume each entry of $K$ is from Gaussian $\mathcal{N}(0, \sigma_k^2)$, and each entry of $Q$ is from Gaussian $\mathcal{N}(0, \sigma_q^2)$. Let $\delta \in (0, 1)$ denote the failure probability. Let $\sigma_a = 4 \cdot (1 + d^{-1} \log(m/\delta))^{1/2} \cdot \sigma_q \sigma_k$. Let $b = \sigma_a \cdot \sqrt{0.4 \log n}$. Let HSR data structure be defined as Part 2 in Corollary 3.5. There exists an algorithm (Algorithm 2), with at least $1 - \delta$ probability, has the following performance:*

- **Part 1.** *The* INIT *procedure runs in* $O(n^{\lfloor d/2 \rfloor})$ *time.*

- **Part 2.** *For each query, the* INFERENCE *procedure runs in* $O(mn^{4/5})$ *time.*

The general framework for full attention computation is quite different from the previous one. Namely, we choose the Part 2 result of the HSR data structure. Since in each inference, both the query matrix $Q$ and the key matrix $K$ differ from any other inference, we first initialize the HSR data structure with the key matrix $K$. Then for each row of the query matrix $Q$, we query the HSR data structure to find the activated entries.

**Lemma 6.3** (General full attention computation framework, informal version of Lemma B.1). *Let* $Q \in \mathbb{R}^{m \times d}$ *and* $K, V \in \mathbb{R}^{n \times d}$ *be defined as Definition 1.2. Assume each entry of $K$ is from Gaussian* $\mathcal{N}(0, \sigma_k^2)$, *and each entry of $Q$ is from Gaussian* $\mathcal{N}(0, \sigma_q^2)$. *Let* $\delta \in (0, 1)$ *denote the failure probability. Let* $\sigma_a = 4 \cdot (1 + d^{-1} \log(m/\delta))^{1/2} \cdot \sigma_q \sigma_k$. *Let* $b = \sigma_a \cdot \sqrt{0.4 \log n}$. *Let* HSR *data structure be defined as Part 1 in Corollary 3.5. There exists an algorithm (Algorithm 3), with at least $1 - \delta$ probability, computes full attention of $Q, K, V$ in* $O(mn^{1-1/\lfloor d/2 \rfloor} + mn^{4/5})$ *time.*

### 6.3 Error Analysis of Softmax Attention with Top-$r$ Indices

Calculating the Softmax attention on the "massive actavted" index set will introduce approximation error. In the following Lemma, we analyze the quantity of this approximation error. Here, we use $\alpha$ to denote the summation of all entries activated by $\exp(x)$ function, and we use $\overline{\alpha}$ to denote the summation of those entries which are excluded from "massive activated" index set. We provide the general error bound of Softmax attention with index set as follows.

**Lemma 6.4** (General error analysis of Softmax attention with index set, informal version of Lemma F.1). *Let* $Q \in \mathbb{R}^{m \times d}$, $K, V \in \mathbb{R}^{n \times d}$ *and the Softmax attention* $\mathsf{Attn}_s$ *be defined in Definition 1.1. Let* $q \in \mathbb{R}^d$ *denote a single row of* $Q \in \mathbb{R}^{m \times d}$. *Let* $\alpha, \overline{\alpha}$ *and* $\widehat{\mathsf{Attn}}_s$ *be defined as Definition 3.2. Then we have* $\|\mathsf{Attn}_s(q, K, V) - \widehat{\mathsf{Attn}}_s(q, K, V)\|_\infty \leq \frac{2\overline{\alpha}}{\alpha} \cdot \|V\|_\infty$.

Note that Lemma 6.4 only provides a general error analysis of Softmax attention with index set. Under mild assumption on the distribution of attention scores, we show that this error is actually very small. For more details, please refer to Theorem 4.3.

## 7 Experiments

In this section, we present our empirical results of evaluating three mainstream LLMs with Softmax attention with top-$r$ indices on different $r$, showing that the results of the experiments are consistent with our theoretical analysis.

**Datasets.** To estimate the approximation error of the Softmax attention with "massive activation" entries, we conduct experiments on the PaulGrahamEssays datasets from LLMTest-NeedleInAHaystack (Kamradt, 2024). Specifically, for each article in the dataset, we first input $2^{15} = 32768$ tokens to the LLMs, then generate $1024$ tokens.

**Metric.** We evaluate the generation quality by the classical perplexity. Perplexity is defined as the exponentiated average negative log-likelihood of a sequence. If we have a tokenized sequence $X = (x_0, x_1, \cdots, x_N)$, then the perplexity of $X$ is: $\mathrm{Perplexity}(X) = \exp(-\frac{1}{N} \sum_{i=1}^{N} \log p_\theta(x_i | x_{<i}))$, where $\log p_\theta(x_i | x_{<i})$ is the log-likelihood of the $i$-th token conditioned on the preceding tokens. Intuitively, it can be thought of as an evaluation of the model's ability to predict uniformly among the set of specified tokens in a corpus. Importantly, the tokenization procedure has a direct impact on a model's perplexity which should be taken into consideration when comparing different models.

**Models.** To demonstrate the generalization of our approximation error bound, we conducted experiments on three mainstream large models: LLaMA 3.1 8B Instruct[2] (Meta, 2024), Mistral Nemo 12B Instruct[3] (MistralAI, 2024), and Phi 3.5 Mini 3.8B Instruct[4] (Abdin et al., 2024).

---

[2] https://huggingface.co/meta-llama/Meta-Llama-3.1-8B-Instruct

[3] https://huggingface.co/mistralai/Mistral-Nemo-Base-2407

[4] https://huggingface.co/microsoft/Phi-3.5-mini-instruct

**Results.** The experiments are conducted on the setting discussed in previous paragraphs. We evaluated the performance of three mainstream LLMs using Softmax attention with top-$r$ indices. In particular, we chose $r$ from the set $\{2^2, 2^4, 2^6, 2^8, 2^{10}, 2^{12}, 2^{15}\}$. As depicted in Figure 2, a significant increase in the perplexity (drop in performance) of LLMs is observed only when $r$ falls below $2^4$. This suggests that the "massive activated" tokens are predominantly found within the top-$2^4$ entries. In comparison to the total of $2^{15}$ entries, the "massive activated" entries constitute a relatively minor fraction. The observed results align with our theoretical analysis, confirming that the approximation error of the Softmax attention mechanism with top-$r$ indices is insignificant for larger values of $r$.

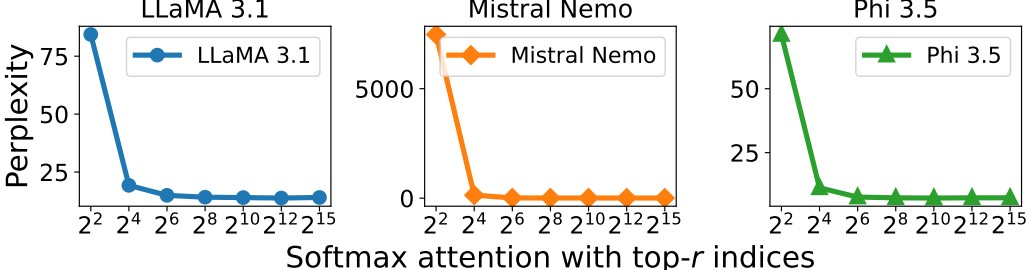

Figure 2: We evaluated the perplexity of three mainstream language models : LLaMA 3.1 8B Instruct, Mistral Nemo 12B, and Phi 3.5 Mini 3.8B Instruct, using Softmax attention with top-$r$ indices on the PaulGrahamEssays dataset. The results indicate a significant increase in perplexity only when the number of selected entries, $r$, falls below $2^4$. This observation aligns with our earlier findings that the proportion of "massive activated" entries is minimal compared to the total number of entries. Furthermore, the approximation error introduced by using top-$r$ indices in Softmax attention remains negligible unless $r$ becomes excessively small.

## 8 DISCUSSION AND FUTURE WORK

The sparsity within neural networks arises primarily from the incorporation of non-linear activation functions. These non-linear functions determine the mechanism or circuit of the neural networks, e.g., the induction head in transformers (Olsson et al., 2022). Gaining insight into these non-linear layers not only enhances our understanding of how neural networks work but also paves the way for optimizing training and inference. We hope our analysis may inspire efficient neural network architecture design. This work represents the initial point of this envisioned blueprint. We concentrate on analyzing the combinations of LLMs and fundamental non-linear activation functions—ReLU and Softmax, which are most relevant to contemporary applications. By analyzing these functions, we aim to demonstrate to the research community that a thorough examination of a model's non-linear characteristics can significantly enhance the running time complexity of neural networks.

In real-world scenarios, a multitude of non-linear activation functions exist beyond ReLU and Softmax, such as those designated as $\mathsf{SELU}(x) = \text{scale} \cdot (\max(0, x) + \min(0, \alpha \cdot (\exp(x) - 1)))$ (Klambauer et al., 2017), $\mathsf{CELU}(x) = \max(0, x) + \min(0, \alpha \cdot (\exp(x/\alpha) - 1))$ (Barron, 2017), and $\mathsf{PRELU}(x) = \max(0, x) + \text{weight} \cdot \min(0, x)$ (He et al., 2015). However, analyzing these alternative functions poses multiple challenges. Hence, we will explore these additional functions in the future.

## 9 CONCLUSION

This work investigates the exploitation of the intrinsic sparsity present in both ReLU and Softmax attention mechanisms to decrease the computational complexity of full attention computation and attention generation scenarios. Specifically, we employ the Half-Space Reporting (HSR) data structure to accelerate the process of identifying non-zero or "massive activated" entries within ReLU and Softmax attentions, respectively. Importantly, our approach does not import any errors to ReLU attention, and it results in only a negligible approximation error for Softmax attention.

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

# Appendix

**Roadmap.** In Section A, we introduce more fundamental lemmas and facts. In Section B, we extend the analysis to ReLU attention calculation, demonstrating improved performance over standard attention computation under specific conditions. In Section C, we first introduce and analyze the time complexity of ReLU attention generation using half-space reporting (HSR) data structures. In Section D, we analyze the sparsity of ReLU attention matrices. In Section E, we introduce our results on reducing the running time of Softmax attention. In Section F, we analyze error bounds for Softmax attention with index sets, balancing efficiency and accuracy.

## A  PRELIMINARY

In this section, we display more fundamental concepts. In Section A.1, we introduce several important probability properties and bounds. In Section A.2, we detail the time complexity and performance of half-space reporting (HSR) data structures.

### A.1  PROBABILITY TOOLS

We state several fundamental properties and bounds for some common distributions.

**Fact A.1** (Weighted summation of Gaussian)**.** *If the following conditions hold:*

- *Let $x \in \mathbb{R}^d$ be a fixed vector and $y \in \mathbb{R}^d$ be a random vector.*

- *For $i \in [d]$, let $x_i$ denote the $i$-th entry of $x$.*

- *Suppose for $i \in [d]$, $y_i \sim \mathcal{N}(0, \sigma^2)$.*

*Then the inner product of $x$ and $y$, $\langle x, y \rangle$ conforms Gaussian distribution $\mathcal{N}(0, \|x\|_2^2 \sigma^2)$. Namely, we have $\langle x, y \rangle \sim \mathcal{N}(0, \|x\|_2^2 \sigma^2)$.*

**Fact A.2** (Independence between $\langle x, y_i \rangle$ and $\langle x, y_j \rangle$)**.** *If the following conditions hold:*

- *Let $x \in \mathbb{R}^d$ be a fixed vector.*

- *Let $y_1, y_2, \cdots y_n \in \mathbb{R}^d$ be $n$ random vectors.*

- *For any $i, j \in [n], i \neq j$, $y_i$ and $y_j$ are independent.*

*Then, for any $i, j \in [n], i \neq j$, $\langle x, y_i \rangle$ and $\langle x, y_j \rangle$ are independent.*

We provide tail bounds for chi-square and Gaussian distributed random variables:

**Lemma A.3** (Chi-square tail bound, Lemma 1 in Laurent & Massart (2000) )**.** *Let $X \sim \mathcal{X}_k^2$ be a chi-squared distributed random variable with $k$ degrees of freedom. Each one has zero means and $\sigma^2$ variance.*

*Then, it holds that*

$$\Pr[X - k\sigma^2 \geq (2\sqrt{kt} + 2t)\sigma^2] \leq \exp(-t)$$
$$\Pr[k\sigma^2 - X \geq 2\sqrt{kt}\sigma^2] \leq \exp(-t)$$

**Fact A.4** (Gaussian tail bound)**.** *Suppose we have a random variable $x \sim \mathcal{N}(\mu, \sigma)$.*

*Then, for $t \in \mathbb{R}$, we have*

$$\Pr[x \geq \mu + t] \leq \exp(-\frac{t^2}{2\sigma^2})$$

*Proof.* We can show

$$\Pr[x \geq \mu + t] = \Pr[x - \mu \geq t]$$
$$= \Pr[e^{x-\mu} \geq e^t]$$

$$= \inf_{\lambda \geq 0} \Pr[e^{\lambda(x-\mu)} \geq e^{\lambda t}]$$

$$\leq \inf_{\lambda \geq 0} \frac{\mathbb{E}[e^{\lambda(x-\mu)}]}{e^{\lambda t}} \tag{1}$$

where the first step, the second step follows from basic algebra, the third step follows from that the inequality holds for any $\lambda > 0$, and the fourth step follows from Markov's inequality.

Then we consider the numerator and we use $y = x - \mu$ to simplify the calculation, we have

$$\mathbb{E}[e^{\lambda y}] = \int_{\mathbb{R}} e^{\lambda y} \frac{e^{-y^2/2\sigma^2}}{\sqrt{2\pi}\sigma} \mathrm{d}y$$

$$= \int_{\mathbb{R}} \frac{e^{-(y-\lambda/\sigma^2)^2 \cdot \frac{1}{2\sigma^2}} e^{\lambda^2 \sigma^2/2}}{\sqrt{2\pi}\sigma} \mathrm{d}y$$

$$= e^{\frac{\lambda^2 \sigma^2}{2}} \int_{\mathbb{R}} \frac{e^{-(y-\lambda/\sigma^2) \cdot \frac{1}{2\sigma^2}}}{\sqrt{2\pi}\sigma} \mathrm{d}y$$

$$= e^{\frac{\lambda^2 \sigma^2}{2}} \tag{2}$$

where the first step follows from the definition of the moment generating function, the second and the third steps follow from basic algebra, and the fourth step follows from the property of the probability density function.

Then we have

$$\Pr[x \geq \mu + t] \leq \inf_{\lambda \geq 0} \exp(\frac{\lambda^2 - \sigma^2}{2} - \lambda t)$$

$$\leq \exp(-\frac{t^2}{2\sigma^2})$$

where the first step follows from Eq. (1) and Eq.(2), the second step follows from the calculation of infimum. □

The Bernstein's inequality for bounding sums of independent random variables is:

**Lemma A.5** (Bernstein inequality Bernstein (1924))**.** *Assume* $Z_1, \cdots, Z_n$ *are* $n$ *i.i.d. random variables.* $\forall i \in [n]$, $\mathbb{E}[Z_i] = 0$ *and* $|Z_i| \leq M$ *almost surely. Let* $Z = \sum_{i=1}^n Z_i$. *Then,*

$$\Pr[Z > t] \leq \exp\left(-\frac{t^2/2}{\sum_{j=1}^n \mathbb{E}[Z_j^2] + Mt/3}\right), \forall t > 0.$$

## A.2 Half-Space Reporting (HSR) Data Structures

The time complexity of the HSR data structure is:

**Theorem A.6** (Agarwal, Eppstein and Matousek Agarwal et al. (1992))**.** *Let* $d$ *be a fixed constant. Let* $t$ *be a parameter between* $n$ *and* $n^{\lfloor d/2 \rfloor}$. *There is a dynamic data structure for half-space reporting that uses* $O_{d,\epsilon}(t^{1+\epsilon})$ *space and pre-processing time,* $O_{d,\epsilon}(\frac{n}{t^{1/\lfloor d/2 \rfloor}} \log n + k)$ *time per query where* $k$ *is the output size and* $\epsilon > 0$ *is any fixed constant, and* $O_{d,\epsilon}(t^{1+\epsilon}/n)$ *amortized update time.*

As a direct corollary, we have

**Corollary A.7** (HSR data-structure time complexity Agarwal et al. (1992), formal version of Corollary 3.5)**.** *Let* $\mathcal{T}_{\mathsf{init}}$ *denote the pre-processing time to build the data structure,* $\mathcal{T}_{\mathsf{query}}$ *denote the time per query, and* $\mathcal{T}_{\mathsf{update}}$ *time per update. Given a set of* $n$ *points in* $\mathbb{R}^d$, *the half-space range reporting problem can be solved with the following performances:*

- *Part 1.* $\mathcal{T}_{\mathsf{init}}(n, d) = O_d(n \log n)$, $\mathcal{T}_{\mathsf{query}}(n, d, k) = O(dn^{1-1/\lfloor d/2 \rfloor} + dk)$.

- *Part 2.* $\mathcal{T}_{\mathsf{init}}(n, d) = O(n^{\lfloor d/2 \rfloor})$, $\mathcal{T}_{\mathsf{query}}(n, d, k) = O(d \log(n) + dk)$.

## B   FULL ReLU ATTENTION COMPUTATION

In this section, we focus on optimizing the standard ReLU attention calculation. By leveraging a HSR data structure and assuming sparsity, the time complexity can be reduced to $O(n^{1+4/5}d)$.

**Lemma B.1** (General full attention computation framework, formal version of Lemma 6.3). *If the following conditions hold:*

- *Let $Q \in \mathbb{R}^{m \times d}$ and $K, V \in \mathbb{R}^{n \times d}$ be defined as Definition 1.2.*

- *Assume each entry of $K$ is from Gaussian $\mathcal{N}(0, \sigma_k^2)$, and each entry of $Q$ is from Gaussian $\mathcal{N}(0, \sigma_q^2)$.*

- *Let $\delta \in (0, 1)$ denote the failure probability.*

- *Let $\sigma_a = 4 \cdot (1 + d^{-1} \log(m/\delta))^{1/2} \cdot \sigma_q \sigma_k$.*

- *Let $b = \sigma_a \cdot \sqrt{0.4 \log n}$.*

- *Let HSR data structure be defined as Part 1 in Corollary A.7.*

*There exists an algorithm (Algorithm 3), with at least $1 - \delta$ probability, computes full attention of $Q, K, V$ in $O(mn^{1-1/\lfloor d/2 \rfloor} + mn^{4/5})$ time.*

*Proof.* For $i \in [m]$, let $\widetilde{k}_i := |\widetilde{S}_{i,\text{fire}}|$ denote the number of non-zero entries in $i$-th row of $A \in \mathbb{R}^{m \times n}$.

The running time for INFERENCE procedure can be written as

$$\mathcal{T}_{\text{init}}(n, d) + \sum_{i=1}^{m} \mathcal{T}_{\text{query}}(n, d, \widetilde{k}_i) + O(d \sum_{i=1}^{m} \widetilde{k}_i) + O(d \sum_{i=1}^{m} \widetilde{k}_i)$$

The first term $\mathcal{T}_{\text{init}}(n, d)$ corresponds to the initialization of the HSR data structure. Since we use Part 1 result from Corollary A.7, the running time for initialization is $\mathcal{T}_{\text{init}}(m, d) = O_d(m \log m)$.

The second term $\sum_{i=1}^{m} \mathcal{T}_{\text{query}}(n, d, \widetilde{k}_i)$ comes from the HSR query operation (Line 11). Since we use Part 1 result from Corollary A.7, we have

$$\sum_{i=1}^{m} \mathcal{T}_{\text{query}}(n, d, \widetilde{k}_i) = O(mn^{1-1/\lfloor d/2 \rfloor} d + d \sum_{i=1}^{m} \widetilde{k}_i)$$
$$= O(mn^{1-1/\lfloor d/2 \rfloor} d + mn^{4/5} d)$$

where the first step follows from $\mathcal{T}_{\text{query}}(n, d, \widetilde{k}_i) = O(dn^{1-\lfloor d/2 \rfloor} + d\widetilde{k}_i)$ (Part 1 of Corollary A.7), the second step follows from with high probability $\widetilde{k}_i$ at most $n^{4/5}$ (Lemma D.3).

The third term $O(\sum_{i=1}^{m} \widetilde{k}_i)$ corresponds to calculating $A_{j,i}$ (Line 13). By Lemma D.3, we have the third term is $O(mn^{4/5})$.

The fourth term $O(\sum_{i=1}^{m} \widetilde{k}_i)$ corresponds to calculating $D^{-1}AV$. Since for $i$-th row of $A$, there are $\widetilde{k}_i$ non-zero entries. Therefore, it takes $O(\sum_{i=1}^{m} \widetilde{k}_i)$ time for calculating $D^{-1}A$. Therefore, it takes $O(d \sum_{i=1}^{m} \widetilde{k}_i)$ time to calculate $D^{-1}AV$. By Lemma D.3, with high probability, $\widetilde{k}_i$ is at most $n^{4/5}$. Therefore, we have the third term as $O(mn^{4/5}d)$.

To sum up, the overall running time is $O(mn^{1-1/\lfloor d/2 \rfloor} d + mn^{4/5} d)$. □

We can now derive a more specific result for the full ReLU attention computation:

**Theorem B.2** (Running time of full ReLU attention computation, formal version of Lemma 5.1). *If the following conditions hold:*

- *Let ReLU attention be defined as Definition 1.2.*

- *Assume each entry of $K$ is from Gaussian $\mathcal{N}(0, \sigma_k^2)$, and each entry of $Q$ is from Gaussian $\mathcal{N}(0, \sigma_q^2)$.*

- *Let $\delta \in (0, 1)$ denote the failure probability.*

- *Let $\sigma_a = 4 \cdot (1 + d^{-1} \log(m/\delta))^{1/2} \cdot \sigma_q \sigma_k$.*

- *Let $b = \sigma_a \cdot \sqrt{0.4 \log n}$.*

- *Suppose we have $Q, K, V \in \mathbb{R}^{n \times d}$.*

*There exists an algorithm (Algorithm 3), with probability at least $1 - \delta$, takes $O(n^{2-1/\lfloor d/2 \rfloor} d + n^{1+4/5} d)$ time to compute the full ReLU attention of $Q, K, V$.*

*Proof.* By Lemma B.1, we have that the FULLATTENTIONCOMPUTATION data structure (Algorithm 3) can run INFERENCE to calculate the ReLU attention, in $O(m^{1-\lfloor d/2 \rfloor} nd + mn^{4/5} d)$ time.

By our assumption, we have $Q \in \mathbb{R}^{n \times d}$. For each calculation, we only need to call FULLATTENTIONCOMPUTATION.INFERENCE$(K, Q, V, n, n, d)$ for once.

Then, we have the ReLU attention calculation run in $O(n^{1+4/5} d)$ time. $\qquad\square$

## C    ReLU Attention Generation

In this section, we present a theoretical analysis of the time complexity of ReLU attention generation using a HSR data structure.

**Lemma C.1** (General attention generation framework, formal version of Lemma 6.2)**.** *If the following conditions hold:*

- *Let $Q \in \mathbb{R}^{m \times d}$ and $K, V \in \mathbb{R}^{n \times d}$ be defined as Definition 1.2.*

- *Assume each entry of $K$ is from Gaussian $\mathcal{N}(0, \sigma_k^2)$, and each entry of $Q$ is from Gaussian $\mathcal{N}(0, \sigma_q^2)$.*

- *Let $\delta \in (0, 1)$ denote the failure probability.*

- *Let $\sigma_a = 4 \cdot (1 + d^{-1} \log(m/\delta))^{1/2} \cdot \sigma_q \sigma_k$.*

- *Let $b = \sigma_a \cdot \sqrt{0.4 \log n}$.*

- *Let HSR data structure be defined as Part 2 in Corollary A.7.*

*Then, there exists an algorithm (Algorithm 2), with at least $1 - \delta$ probability, has the following performance:*

- **Part 1.** *The INIT procedure runs in $O(n^{\lfloor d/2 \rfloor})$ time.*

- **Part 2.** *For each query, the INFERENCE procedure runs in $O(mn^{4/5} d)$ time.*

*Proof.* **Proof of Part 1.**

The INIT procedure only runs the initialization of the HSR data structure. Since we use Part 2 result from Corollary A.7, the running time of INIT procedure is $\mathcal{T}_{\mathsf{init}}(n, d) = O(n^{\lfloor d/2 \rfloor})$.

**Proof of Part 2.**

For $i \in [m]$, let $\widetilde{k}_i := |\widetilde{S}_{i,\mathrm{fire}}|$ denote the number of non-zero entries in $i$-th row of $A \in \mathbb{R}^{m \times n}$.

The running time for INFERENCE procedure can be written as

$$\sum_{i=1}^{m} \mathcal{T}_{\mathsf{query}}(n, d, \widetilde{k}_i) + O(d \sum_{i=1}^{m} \widetilde{k}_i) + O(d \sum_{i=1}^{m} \widetilde{k}_i)$$

The first term $\sum_{i=1}^{m} \mathcal{T}_{\text{query}}(n, d, \widetilde{k}_i)$ corresponds to the HSR query operation (Line 16). Since we use the Part 2 result from Corollary A.7, we have

$$\sum_{i=1}^{m} \mathcal{T}_{\text{query}}(n, d, \widetilde{k}_i) = O(md \log n + d \sum_{i=1}^{m} \widetilde{k}_i)$$
$$= O(md \log n + mn^{4/5}d)$$
$$= O(mn^{4/5}d)$$

where the first step follows from $\mathcal{T}_{\text{query}}(n, d, k) = O(d \log n + dk)$ in Part 2 of Corollary A.7, the second step follows from with high probability, $\widetilde{k}_i$ is at most $n^{4/5}$ (Lemma D.3), the third step follows from $\log n < n^{4/5}$.

The second term $O(d \sum_{i=1}^{m} \widetilde{k}_i)$ corresponds to calculating $A_{i,j}$ (Line 18). There are $m$ iterations, and in each iteration, it calculates $\widetilde{k}_i$ entries of $A$. Then, the second term is $O(d \sum_{i=1}^{m} \widetilde{k}_i)$. By Lemma D.3, with high probability, $\widetilde{k}_i$ is at most $n^{4/5}$. Therefore, we have the second term as $O(mn^{4/5}d)$.

Similar to the proof of Lemma B.1 this term is $O(mn^{4/5}d)$.

To sum up, we have the overall running time for INFERENCE procedure is $O(mn^{4/5}d)$. $\qquad\square$

We now derive a comprehensive sparsity analysis for the ReLU attention mechanism:

**Theorem C.2** (Running time of full ReLU attention generation, formal version of Theorem 4.1)**.** *If the following conditions hold:*

- *Let ReLU attention be defined as Definition 1.2.*

- *Assume each entry of $K$ is from Gaussian $\mathcal{N}(0, \sigma_k^2)$, and each entry of $Q$ is from Gaussian $\mathcal{N}(0, \sigma_q^2)$.*

- *Let $\delta \in (0, 1)$ denote the failure probability.*

- *Let $\sigma_a = 4 \cdot (1 + d^{-1} \log(m/\delta))^{1/2} \cdot \sigma_q \sigma_k$.*

- *Let $b = \sigma_a \cdot \sqrt{0.4 \log n}$.*

- *Suppose we have KV Cache $K, V \in \mathbb{R}^{n \times d}$. We want to generate a $m$ length answer, where $n \gg m$.*

*There exists an algorithm (Algorithm 2), with at least $1 - \delta$ probability, takes $O(mn^{4/5}d)$ time to generate the answer.*

*Proof.* We make use of the ATTENTIONGENERATION data structure (Algorithm 2) in Lemma C.1.

The generation process is an auto-regressive procedure, we define the following notations for better understanding. For $i \in [m]$, let $q_i, k_i \in \mathbb{R}^d$ denote the query vector of the $i$-th iteration, respectively. Note that $q_i$ need to attend on both $K \in \mathbb{R}^{n \times d}$ and $\{k_1, k_2, \cdots, k_{i-1}\}$.

For calculating the attention between $q_i$ and $K \in \mathbb{R}^{n \times d}$, we just need to call ATTENTIONGENERATION .INFERENCE($q_i, 1$) for once. Therefore the running time for this part is $O(n^{4/5}d)$ time.

For calculating the attention between $q_i$ and $\{k_1, k_2, \cdots, k_{i-1}, k_i\}$, it takes $O(i \cdot d)$ time.

Therefore, for a single query $q_i$, the running time for getting the attention matrix $A \in \mathbb{R}^{1 \times (n+i)}$ is $(n^{4/5} + i) \cdot d$. Since there are only $n^{4/5} + i$ non-zero entries in $A$, it takes $n^{4/5} + i$ time to calculate $D^{-1}A$. Then, it takes $(n^{4/5} + i) \cdot d$ time to calculate $D^{-1}AV$. Since $i \leq m$, the total running time for calculating attention for a single query $q_i$ is $O((n^{4/5} + m) \cdot d)$.

There are $m$ queries in total. The running time for $m$ queries is $O(mn^{4/5}d + m^2d)$.

Since we have $n \gg m$, the overall running time for the generation is $O(mn^{4/5}d)$. $\qquad\square$

# D   SPARSITY ANALYSIS

To begin our analysis, we first examine the application of Bernstein's inequality to the matrix $K$:

**Lemma D.1** (Bernstein on $K$). *If the following conditions hold:*

- *Let the ReLU attention be defined as Definition 1.2.*

- *Let $Q \in \mathbb{R}^{m \times d}$ and $K, V \in \mathbb{R}^{n \times d}$ be defined as Definition 1.2.*

- *Let $b \in \mathbb{R}$ denote the threshold of ReLU activation, as defined in Definition 1.2.*

- *For $i \in [m]$, let $\widetilde{k}_i$ denote the number of non-zero entries in $i$-th row of $A \in \mathbb{R}^{m \times n}$.*

- *Assume each entry of $K$ is from Gaussian $\mathcal{N}(0, \sigma_k^2)$*

- *Let $x \in \mathbb{R}^d$ denote a single row of $Q \in \mathbb{R}^{m \times d}$.*

- *Let $\sigma_a = \|x\|_2 \sigma_k / \sqrt{d}$.*

*Then, we can show that, with probability at least $1 - \exp(-\Omega(n \cdot \exp(-\frac{b^2}{2\sigma_a^2})))$, the number of non-zero entries $\widetilde{k}_i$ is at most $2n \cdot \exp(-\frac{b^2}{2\sigma_a^2})$. Namely, we have*

$$\Pr[\widetilde{k}_i \leq 2n \cdot \exp(-\frac{b^2}{2\sigma_a^2})] \geq 1 - \exp(-\Omega(n \cdot \exp(-\frac{b^2}{2\sigma_a^2})))$$

*Proof.* For simplicity, for $i \in [n], j \in [d]$, we use $K_{i,j} \in \mathbb{R}$ to denote the $(i,j)$-th entry of $K \in \mathbb{R}^{n \times d}$.

Let $r_i \in \{0, 1\}$ be the indicator function of $\langle x, K_{i,*} \rangle$. Then, we have $\widetilde{k}_i = \sum_{j=1}^{n} r_j$.

Since $r_i$ is an indicator function, then we have

$$|r_i| \leq 1.$$

By assumption, we have $K_{i,j} \sim \mathcal{N}(0, \sigma_k^2)$.

Let $\sigma_a = \|x\|_2 \cdot \sigma_k / \sqrt{d}$.

By the property of Gaussian distribution (Fact A.1), we have $\langle x, K_{i,*} \rangle \sim \mathcal{N}(0, d \cdot \sigma_a^2)$ and $\langle x, K_{i,*} \rangle / \sqrt{d} \sim \mathcal{N}(0, \sigma_a^2)$.

For any $i, j \in [n]$, by Fact A.2, we have $\langle x, K_{i,*} \rangle$ and $\langle x, K_{j,*} \rangle$ are independent, which implies $r_i$ and $r_j$ are independent.

By the tail bound of Gaussian distribution (Fact A.4), we have

$$\Pr[r_i = 1] = \Pr[\langle x, K_{i,*} \rangle / \sqrt{d} \geq b]$$
$$\leq \exp(-\frac{b^2}{2\sigma_a^2}),$$

which implies

$$\mathbb{E}[r_i] \leq \exp(-\frac{b^2}{2\sigma_a^2}), \tag{3}$$

and

$$\mathbb{E}[r_i^2] \leq \exp(-\frac{b^2}{2\sigma_a^2}),$$

which implies

$$\sum_{i=1}^{n} \mathbb{E}[r_i^2] \leq n \cdot \exp(-\frac{b^2}{2\sigma_a^2}).$$

Since we have $\widetilde{k}_i = \sum_{j=1}^{n} r_j$, by Eq. (3), we have

$$E[\widetilde{k}_i] \leq n \cdot \exp(-\frac{b^2}{2\sigma_a^2}).$$

Let $k_0 := n \cdot \exp(-\frac{b^2}{2\sigma_a^2})$. By the Bernstein inequality (Lemma A.5), we have

$$\Pr[\widetilde{k}_i \geq k_0 + t] \leq \exp(-\frac{t^2/2}{k_0 + t/3}) \tag{4}$$

We choose $t = k_0$, then we have

$$\Pr[\widetilde{k}_i \geq 2k_0] \leq \exp(-3k_0/8)$$

Then, we reach our conclusion: with probability at least $1 - \exp(-\Omega(n \cdot \exp(-\frac{b^2}{2\sigma_a^2})))$, the number of non-zero entries in each row of the attention matrix $A$ is bounded by $\widetilde{k}_i \leq 2n \cdot \exp(-\frac{b^2}{2\sigma_a^2})$.

$\square$

We turn our attention to bounding $\|x\|_2$:

**Lemma D.2** ($\|x\|_2$ bound). *If the following conditions hold:*

- *Let $Q \in \mathbb{R}^{m \times d}$ be defined as Definition 1.2.*
- *Let $x \in \mathbb{R}^d$ denote a single row of $Q \in \mathbb{R}^{m \times d}$.*
- *Assume each entry of $Q$ is from $\mathcal{N}(0, \sigma_q^2)$.*

*Then, we can show that, for $t \geq 0$ with probability $1 - \exp(-t)$, $\|x\|_2$ is at most $\sqrt{3} \cdot (d+t)^{1/2} \cdot \sigma_q$. Namely, we have*

$$\Pr[\|x\|_2 \leq \sqrt{3} \cdot (d+t)^{1/2} \cdot \sigma_q] \geq 1 - \exp(-t).$$

*Proof.* For simplicity, we use $x_i \in \mathbb{R}$ to denote the $i$-th entry of $x$.

By the assumption, we have $x_i \sim \mathcal{N}(0, \sigma_q^2)$.

Since $\|x\|_2^2 = \sum_{i=1}^{d} x_i^2$, by Chi-square tail bound (Lemma A.3), we have

$$\Pr[\|x\|_2^2 - d\sigma_q^2 \geq (2\sqrt{dt} + 2t)\sigma_q^2] \leq \exp(-t),$$

which implies

$$\Pr[\|x\|_2^2 \geq (2\sqrt{dt} + 2t + d)\sigma_q^2] \leq \exp(-t). \tag{5}$$

Since we have $2\sqrt{dt} \leq d + t$, Eq. (5) implies

$$\Pr[\|x\|_2^2 \geq 3(d+t)\sigma_q^2] \leq \exp(-t),$$

which is equivalent to

$$\Pr[\|x\|_2 \geq \sqrt{3} \cdot (d+t)^{1/2} \cdot \sigma_q] \leq \exp(-t).$$

$\square$

We can now present our formal sparsity analysis, which builds upon the previous lemmas:

**Lemma D.3** (Sparsity analysis, formal version of Lemma 6.1). *If the following conditions hold:*

- *Let the ReLU attention be defined as Definition 1.2.*

- *Let $Q \in \mathbb{R}^{m \times d}$ and $K, V \in \mathbb{R}^{n \times d}$ be defined as Definition 1.2.*

- *Let $b \in \mathbb{R}$ denote the threshold of ReLU activation, as defined in Definition 1.2.*

- *For $i \in [m]$, let $\widetilde{k}_i$ denote the number of non-zero entries in $i$-th row of $A \in \mathbb{R}^{m \times n}$.*

- *Assume each entry of $K$ is from Gaussian $\mathcal{N}(0, \sigma_k^2)$, and each entry of $K$ is from Gaussian $\mathcal{N}(0, \sigma_q^2)$.*

- *Let $\delta \in (0, 1)$ denote the failure probability.*

- *Let $\sigma_a = 4 \cdot (1 + d^{-1} \log(m/\delta))^{1/2} \cdot \sigma_q \sigma_k$.*

- *Let $b = \sigma_a \cdot \sqrt{0.4 \log n}$.*

*Then, we can show that, with probability at least $1 - \delta$, for all $i \in [m]$, the number of non-zero entries of the $i$-th row $\widetilde{k}_i$ is at most $2n^{4/5}$.*

*Proof.* This proof follows from applying union bound on Lemma D.1 and Lemma D.2.

By Lemma D.2, we have

$$\Pr[\|x\|_2 \leq \sqrt{3} \cdot (d + t)^{1/2} \cdot \sigma_q] \geq 1 - \exp(-t). \tag{6}$$

We choose $t = d + \log(m/\delta)$. Then, Eq. (6) implies

$$\Pr[\|x\|_2 \leq 4 \cdot (d + \log(m/\delta))^{1/2} \cdot \sigma_q] \geq 1 - \exp(-(d + \log(m/\delta))). \tag{7}$$

Let $\sigma_a = \|x\|_2 \cdot \sigma_k / \sqrt{d}$. By Eq.(7), we have $\sigma_a = 4 \cdot (1 + d^{-1} \log(m/\delta))^{1/2} \cdot \sigma_q \sigma_k$.

By Lemma D.1, we have

$$\Pr[\widetilde{k}_i \leq 2n \cdot \exp(-\frac{b^2}{2\sigma_a^2})] \geq 1 - \exp(-\Omega(n \cdot \exp(-\frac{b^2}{2\sigma_a^2}))). \tag{8}$$

Let $b = \sigma_a \cdot \sqrt{0.4 \log n}$. Then, Eq. (8) implies

$$\Pr[\widetilde{k}_i \leq 2n^{4/5}] \geq 1 - \exp(-O(n^{4/5})) \tag{9}$$

Since we have $n \gg d$, this implies

$$\exp(-O(n^{4/5})) \leq \exp(-d) \tag{10}$$

Taking union bound over Eq. (7) and Eq. (9), we have

$$\Pr[\widetilde{k}_i \leq 2n^{4/5}] \geq 1 - (\exp(-O(n^{4/5}) + \exp(-(d + \log(m/\delta))))$$
$$= 1 - (\exp(-O(n^{4/5}) + (\delta/m) \cdot \exp(-d)))$$
$$\geq 1 - \delta/m. \tag{11}$$

where the first step follows from the union bound, the second step follows from basic algebra, the third step follows from Eq. (10).

Since $x \in \mathbb{R}$ represents a single row of $Q \in \mathbb{R}^{m \times d}$, we already proved that for each fixed row of $A$, the $\widetilde{k}_i$ is at most $2n^{4/5}$ with probability $1 - \delta/m$.

Taking the union bound over $m$ rows in $A$, then we can show that with probability $1 - \delta$, for all rows of $A$, that row's $\widetilde{k}_i$ is at most $2n^{4/5}$.

$\square$

## E   Running Time of Softmax Attention

In this section, we provide our results on reducing the running time of Softmax attention. We begin with introducing our result on Softmax attention generation.

**Theorem E.1** (Running time of Softmax attention generation, formal version of Theorem 4.2). *Let $Q \in \mathbb{R}^{m \times d}$, $K, V \in \mathbb{R}^{n \times d}$ and the Softmax attention $\mathsf{Attn}_s$ be defined in Definition 1.1. Let $\mathsf{NN}(r, q, K) \subseteq [n]$ and the Softmax attention with index set $\widehat{\mathsf{Attn}}_s$ be defined as Definition 3.2. We choose the threshold $b \in \mathbb{R}$ in Algorithm 2 such that $R = \mathsf{NN}(n^{4/5}, q, K)$. Then, we can show that the Softmax attention with index set $\widehat{\mathsf{Attn}}_s$ achieves outstanding running time under the Softmax attention generation scenario: Suppose we have KV Cache $K, V \in \mathbb{R}^{n \times d}$. We want to generate a $m$ length answer, where $m = \Theta(1)$. Algorithm 2 (replacing ReLU attention with Softmax attention) takes $O(mn^{4/5})$ time to generate the answer.*

*Proof.* The Softmax attention generation scenario can be proved by substituting the ReLU attention $\mathsf{Attn}_r$ (Definition 1.2) with Softmax attention with index set $\widehat{\mathsf{Attn}}_s$ (Definition 3.2) in Algorithm 2 and Theorem 4.1. □

Then, we move on to our result on Softmax full attention computation.

**Theorem E.2** (Running time of Softmax full attention computation, formal version of Theorem 5.2). *Let $Q \in \mathbb{R}^{m \times d}$, $K, V \in \mathbb{R}^{n \times d}$ and the Softmax attention $\mathsf{Attn}_s$ be defined in Definition 1.1. Let $\mathsf{NN}(r, q, K) \subseteq [n]$ and the Softmax attention with index set $\widehat{\mathsf{Attn}}_s$ be defined as Definition 3.2. We choose the threshold $b \in \mathbb{R}$ in Algorithm 3 such that $R = \mathsf{NN}(n^{4/5}, q, K)$. Then, we can show that the Softmax attention with index set $\widehat{\mathsf{Attn}}_s$ achieves outstanding running time under full Softmax attention computation scenario: Suppose we have $m = \Theta(n)$. Algorithm 3 (replacing ReLU attention with Softmax attention) takes $O(n^{2-1/\lfloor d/2 \rfloor}d + n^{1+4/5}d)$ time to calculate the attention output.*

*Proof.* The Softmax full attention computation scenario can be proved by substituting the ReLU attention $\mathsf{Attn}_r$ (Definition 1.2) with Softmax attention with index set $\widehat{\mathsf{Attn}}_s$ (Definition 3.2) in Algorithm 3 and Theorem 5.1. □

## F   Error Analysis of Softmax Attention

In this section, we provide an error analysis of the Softmax attention mechanism, deriving error bounds for the general case and a specific case with the massive activation property.

The following lemmas establish error bounds for Softmax attention when using index sets, formalizing the approximation error in attention computation.

**Lemma F.1** ( General error analysis of Softmax attention with index set, formal version of Lemma 6.4 ). *If the following conditions hold:*

- *Let $Q \in \mathbb{R}^{m \times d}$, $K, V \in \mathbb{R}^{n \times d}$ and the Softmax attention $\mathsf{Attn}_s$ be defined in Definition 1.1.*

- *Let $q \in \mathbb{R}^d$ denote a single row of $Q \in \mathbb{R}^{m \times d}$.*

- *Let $\alpha, \overline{\alpha}$ and $\widehat{\mathsf{Attn}}_s$ be defined as Definition 3.2.*

*Then we have*

$$\|\mathsf{Attn}_s(q, K, V) - \widehat{\mathsf{Attn}}_s(q, K, V)\|_\infty \leq \frac{2\overline{\alpha}}{\alpha} \cdot \|V\|_\infty.$$

*Proof.* Recall that $\overline{R} = [n] \setminus R$ and $\widehat{K} = K_R \in \mathbb{R}^{r \times d}$ and $\widehat{V} = V_R \in \mathbb{R}^{r \times d}$ and $\overline{K} = K_{\overline{R}} \in \mathbb{R}^{(n-r) \times d}$ and $\overline{V} = V_{\overline{R}} \in \mathbb{R}^{(n-r) \times d}$ as defined in Definition 3.1. Also, we have $\widehat{u} = \exp(q\widehat{K}^\top) \in \mathbb{R}^r$ and $\widehat{\alpha} = \langle \widehat{u}, \mathbf{1}_r \rangle \in \mathbb{R}$ and $\overline{u} = \exp(q\overline{K}^\top) \in \mathbb{R}^{n-r}$ and $\overline{\alpha} = \langle \overline{u}, \mathbf{1}_{n-r} \rangle \in \mathbb{R}$ as defined in Definition 3.2.

Then, we have

$$\|\mathsf{Attn}_s(q, K, V) - \widehat{\mathsf{Attn}}_s(q, K, V)\|_\infty$$
$$= \|(\widehat{\alpha} + \overline{\alpha})^{-1}(\widehat{u}\widehat{V} + \overline{u}\overline{V}) - \widehat{\alpha}^{-1}\widehat{u}\widehat{V}\|_\infty$$
$$\leq \|((\widehat{\alpha} + \overline{\alpha})^{-1} - \widehat{\alpha}^{-1})\widehat{u}\widehat{V}\|_\infty + \|(\widehat{\alpha} + \overline{\alpha})^{-1}\overline{u}\overline{V}\|_\infty$$
$$\leq |(\widehat{\alpha} + \overline{\alpha})^{-1} - \widehat{\alpha}^{-1}| \cdot \|\widehat{u}\|_1 \cdot \|\widehat{V}\|_\infty + (\widehat{\alpha} + \overline{\alpha})^{-1} \cdot \|\overline{u}\|_1 \cdot \|\overline{V}\|_\infty$$
$$= (\widehat{\alpha}^{-1} - (\widehat{\alpha} + \overline{\alpha})^{-1}) \cdot \widehat{\alpha} \cdot \|\widehat{V}\|_\infty + (\widehat{\alpha} + \overline{\alpha})^{-1} \cdot \overline{\alpha} \cdot \|\overline{V}\|_\infty$$
$$\leq (\widehat{\alpha}^{-1} - (\widehat{\alpha} + \overline{\alpha})^{-1}) \cdot \widehat{\alpha} \cdot \|V\|_\infty + (\widehat{\alpha} + \overline{\alpha})^{-1} \cdot \overline{\alpha} \cdot \|V\|_\infty$$
$$= 2(\widehat{\alpha} + \overline{\alpha})^{-1} \cdot \overline{\alpha} \cdot \|V\|_\infty$$
$$= 2\alpha^{-1} \cdot \overline{\alpha} \cdot \|V\|_\infty,$$

where the first step is by Definition 3.2, the second step is by triangle inequality, the third step is by $\|uV\|_\infty \leq \|u\|_1 \cdot \|V\|_\infty$ for any vector $u$ and conformable matrix $V$, and the fourth step is by definition of $\widehat{\alpha}$ and $\overline{\alpha}$, i.e., $\widehat{\alpha} = \langle \widehat{u}, \mathbf{1}_r \rangle = \|\widehat{u}\|_1$ (note that each entry of $\widehat{u}$ is positive), the fifth step is by $\max\{\|\widehat{V}\|_\infty, \|\overline{V}\|_\infty\} = \|V\|_\infty$, the sixth step in by simple calculation and the last step is by $\widehat{\alpha} + \overline{\alpha} = \alpha$. □

Building on this, we now present a more specific error analysis incorporating the massive activation property:

**Theorem F.2** (Error analysis of Softmax attention with index set, formal version of Theorem 4.3)**.**
*If the following conditions hold:*

- *Let $Q \in \mathbb{R}^{m \times d}$, $K, V \in \mathbb{R}^{n \times d}$ and the Softmax attention $\mathsf{Attn}_s$ be defined in Definition 1.1.*

- *Let $q \in \mathbb{R}^d$ denote a single row of $Q \in \mathbb{R}^{m \times d}$.*

- *Let $\gamma \in [0, 1]$, $\beta_1 \geq \beta_2 \geq 0$.*

- *Let the Softmax attention with index set $\widehat{\mathsf{Attn}}_s$ be defined as Definition 3.2.*

- *Let $\mathsf{NN}(r, q, K) \subseteq [n]$ denote the indices of top-$r$ entries of $qK$.*

- *Let $R = \mathsf{NN}(n^\gamma, q, K) \subseteq [n]$, where $|R| = n^\gamma$.*

- *Assume the query $q$ and key cache $K$ have $(\gamma, \beta_1, \beta_2)$ massive activation property.*

*Then, we can show that*

$$\|\widehat{\mathsf{Attn}}_s(q, K, V) - \mathsf{Attn}_s(q, K, V)\|_\infty \leq \frac{2\|V\|_\infty}{n^{\gamma + (\beta_1 - \beta_2) \cdot \|q\|_2 - 1}}.$$

*Proof.* Let $\alpha, \overline{\alpha}, \widehat{\alpha}$ be defined in Definition 3.2. By Lemma F.1, we have

$$\|\mathsf{Attn}_s(q, K, V) - \widehat{\mathsf{Attn}}_s(q, K, V)\|_\infty \leq \frac{2\overline{\alpha}}{\alpha} \cdot \|V\|_\infty.$$

By Definition 3.3, we have

$$\widehat{\alpha} = \sum_{i \in \mathsf{NN}(n^\gamma, q, K)} \exp(\langle q, K_i \rangle)$$
$$\geq \sum_{i \in \mathsf{NN}(n^\gamma, q, K)} \exp(\|q\|_2 \beta_1 \log(n))$$
$$= n^{\gamma + \beta_1 \cdot \|q\|_2},$$

where the first step is by Definition of $\widehat{\alpha}$, the second step is by Definition 3.3 and Jensen inequality, and the last step is by simple calculation.

We also have

$$\overline{\alpha} = \sum_{i \in [n] \setminus \mathsf{NN}(n^\gamma, q, K)} \exp(\langle q, K_i \rangle)$$

$$\leq \sum_{i \in [n] \setminus \mathsf{NN}(n^\gamma, q, K)} \exp(\|q\|_2 \beta_2 \log(n))$$

$$\leq n^{1 + \beta_2 \cdot \|q\|_2},$$

where the first step is by Definition of $\overline{\alpha}$, the second step is by Definition 3.3, and the last step is by simple calculation.

Finally, we finish the proof by the fact $\widehat{\alpha} + \overline{\alpha} = \alpha$. $\qquad\square$

