# OpenReview forum: "HSR-Enhanced Sparse Attention Acceleration"
_ICLR.cc/2025/Conference — ICLR 2025 Conference Withdrawn Submission_

### Official Review · Reviewer_b8bx · 2024-10-30

**Soundness:** 3
**Presentation:** 2
**Contribution:** 3
**Rating:** 3
**Confidence:** 2

**Summary:**

Using HSR (half space reporting) data structure, we can detect the non-zero entries in ReLU attention and Softmax attention (treat the near-zero entry as zero by thresholding) in O(mn^0.8) where n is context length, and m is query length.

**Strengths:**

Using the HSR data structure to detect non-zero entries in the attention mechanism is novel and interesting.
Since it can detect exact non-zero entries, we can reconstruct ReLU attention and show negligible errors (as far as the author claims, but I have concerns about the evaluation) in softmax attention.

**Weaknesses:**

1. Lack of proper downstream task evaluation.
 - I do not think the perplexity evaluation shows real-world performance. Also, the Y-axis range of Figure 2 is way too large while considering a 0.1 difference in perplexity is significant in downstream tasks. I suggest to evaluate the method in InfiniteBench (https://github.com/OpenBMB/InfiniteBench) or something more realistic.
2. Lack of latency evaluation (Latency improvement claim may be marginal)
 - Can you provide wall-clock latency in GPU? I do not think the **0.8 speedup is significant speedup, considering this method added a complex algorithm to the top of attention. I think the additional algorithms to reduce theoretical complexity might be more inefficient than FlashAttention (https://github.com/Dao-AILab/flash-attention, https://flashinfer.ai/) or similar works.
 - Can you provide the theoretically required FLOPs or the number of instructions in the GPU?
 Can you discuss parallelism with the HSR data structure? I cannot understand the HSR data structure well with only this paper's description. I am willing to discuss this with the author.

**Questions:**

1. Can you show wall-clock time latency speedup? Typically, the complex data structure decreases theoretic latency but usually increases real-world latency due to overheads or lack of parallelism
2. Can you add some figures to understand the overview intuitively? I think the paper presentation is quite hard to follow. For example, why do we need to define m=(-)(1) and m=(-)(n)? We typically say they are the decoding and prefill (or prompt) phases.

---

Overall, I think the introduction to the HSR data structure is interesting and impactful if the author's claim is true: huge latency speedup in a long context. However, I have concerns that this paper is not carefully investigated about HW and ignores the limitations of real-world hardware such as GPUs. Also, the overall writing is quite hard to follow and needs improvements. Therefore, I think this paper needs significant revisions by including additional analysis and discussions. I hope we can discuss the above problems and improve the paper during the discussion period.

---

### Official Review · Reviewer_bh6x · 2024-10-31

**Soundness:** 2
**Presentation:** 1
**Contribution:** 2
**Rating:** 3
**Confidence:** 3

**Summary:**

This paper presents a novel approach to enhance the efficiency of attention mechanisms in Large Language Models (LLMs), specifically targeting long-context scenarios. By utilizing Half-Space Reporting (HSR), the authors aim to identify non-zero or "massively activated" entries in the attention matrix, reducing computational complexity.

**Strengths:**

1. Addressing a Timely and Important Problem. The paper tackles the critical issue of optimizing sparse attention mechanisms in Large Language Models (LLMs), specifically focusing on both ReLU and Softmax attention sparsity.
2. Leveraging the HSR Data Structure. By leveraging the Half-Space Reporting (HSR) data structure, the paper reduces computational complexity in sparse attention and activation.
3. Theoretical Analysis. It provides rigorous theoretical proofs, ensuring the proposed methods are grounded in solid mathematical foundations.

**Weaknesses:**

1. Very Limited evaluation and lack of comparisons.
2. No cost and quatitative analysis of HSR.

see my questions for details.

**Questions:**

The paper provides a very limited evaluation and does not compare its methods to recent work on attention or ReLU sparsity. This absence of benchmarking against existing approaches makes it challenging to assess the relative effectiveness of the proposed methods.

A lot of qualitive analysis, but no quantitaive results. (if n = 32k, the sparsity ratio is roughly 7/8, is this correct?)
with 8k,16k,32k, or even to 128k context length, how much sparsity can be leveraged? what real speedup can be achieved? what is the overhead of HSR? how does this approach impact model accuracy, such as perplexity or performance on benchmarks like LongBench/Ruler/etc. Can you compare it with recent related works, like StreamingLLM, MInference, etc.

---

### Official Review · Reviewer_bF6Z · 2024-10-31

**Soundness:** 2
**Presentation:** 2
**Contribution:** 2
**Rating:** 3
**Confidence:** 4

**Summary:**

This paper proposes using a Half-Space Reporting data structure to rapid identify non-zero or "massively activated" entries in the attention matrix and hence speed up the computation of attention. The authors prove theoretically that the proposed method achieves a running time of $O(mm^{4/5})$ when the context length is $n$ and the query length is $m$ in the generation phase of LLM, and the running time is $O(mn^{1-1/\lfloor d/2\rfloor} + m n^{4/5})$ for full attention. The method is accurate for ReLU attention and has small errors for softmax attention.

**Strengths:**

1. The complexity is smaller than the original $O(mn)$.

2. The method is accurate for ReLU attention.

**Weaknesses:**

1. The presentation is not clear. For example, it is hard to see the difference between Algorithm 2 and Algorithm 3.

2. The assumptions in the theoretical results are not justified. They may not be relevant in practical settings.

3. No experimental results to demonstrate that the proposed method can speed up the computation of attention in practice

**Questions:**

1.  Algorithm 2 and Algorithm 3 look essentially the same. What's the key difference between them? Is there any intuitive explanation about why the complexities are different in the two settings?

2. The dimension $d$ is typically large, so the theoretical complexity for full attention is very close to $O(mn)$. How large should $n$ be for the proposed algorithm to have any benefit?

3. Is there any justification for the assumptions in the theoretical results? Are they relevant in practice?

4. Can the proposed method be parallelized?

5. Can the proposed method provide any speedup in practice?

---

### Official Review · Reviewer_Sstt · 2024-11-01

**Soundness:** 2
**Presentation:** 2
**Contribution:** 2
**Rating:** 3
**Confidence:** 4

**Summary:**

This paper addresses the computational challenge of attention mechanisms in LLMs with long-context processing. It proposes a novel acceleration technique based on the Half-Space Reporting (HSR) data structure to enhance both Softmax and ReLU attention mechanisms. The proposed method reduces computation for ReLU and Softmax attention with pre-computed key-value (KV) caches in text generation and further improves the time complexity for full attention computation. The approach provides theoretical guarantees, negligible approximation errors for Softmax, and zero error for ReLU attention, supported by empirical evaluations on major LLMs.

**Strengths:**

1. The integration of HSR for sparse attention significantly contributes to reducing computational costs in attention mechanisms.
2. The paper rigorously proves the effectiveness of the approach, including detailed bounds for approximation errors and sparse matrix management.

**Weaknesses:**

1. While the theoretical basis is strong, the paper does not fully explore the practicality of implementing HSR across various LLMs, especially in comparison with other baseline methods on a wider range of benchmarks.
2. The empirical results lack details on latency and memory usage in LLM settings, which are crucial for assessing real-world efficiency.
3. The absence of accessible code for implementation makes it challenging to independently verify the method’s performance.

**Questions:**

Please see the weaknesses.

---

### Note · Authors · 2024-11-14

I have read and agree with the venue's withdrawal policy on behalf of myself and my co-authors.